# Green Supply Chain Decisions Considering Consumers' Low-Carbon Awareness under Different Government Subsidies

**Chang Su [1], Xiaojing Liu [2,\*] and Wenyi Du [3,4]**

[1]   School of Finance and Economics, Yangtze Normal University, Chongqing 408100, China; suchang@yznu.edu.cn

[2]   Business School, Jiangsu Normal University, Xuzhou 221116, China

[3]   Business School, Hohai University, Nanjing 211100, China; wydu@jsnu.edu.cn

[4]   Management Science Institute, Hohai University, Nanjing 211100, China

\*   Correspondence: xjliu@jsnu.edu.cn; Tel.: +86-152-6219-8583

**Abstract:** This study examined how to arrange the generation and pricing of supply chain members in the case of consumer green preference with different government subsidies. The green supply chain comprises a manufacturer and a retailer; the government subsidizes manufacturers who produce green products and consumers who buy green products. The study built a green supply chain pricing decision model with different forms of subsidy under various power structures. By backward induction and sensitivity analysis, this study analyzed optimal strategies of green supply chain under various modes, and we discuss how the government subsidy coefficient affects the optimal decision of a green supply chain. The results show that, firstly, whether the government subsidizes the manufacturers or the consumers, the wholesale price offered by the manufacturer is directly proportional to the subsidy coefficient under the two power structures. Secondly, when the government subsidizes the manufacturer, the carbon-emission level and the retail price are inversely proportional to the subsidy coefficient under the manufacturer leader; the carbon-emission level and the retail price are all directly proportional to the subsidy coefficient under the retailer leader. Finally, when the government subsidizes the consumers, the carbon-emission level and the retail price are directly proportional to the subsidy coefficient under the two power structures.

**Keywords:** green supply chain; government subsidies; consumers' low-carbon behavior; power structure; Stackelberg game

## 1. Introduction

In the process of global industrialization, enterprises are faced with a complex social environment, unpredictable market demand, and environmental protection and sustainable development requirements. Enterprises tend to waste too many resources in the process of manufacturing products, resulting in ecological imbalance, causing global warming and other climate problems, and a damaged ecological environment affects people's lifestyle and consumption behavior [1]. According to a survey for the UK Carbon Trust, more than two-thirds of people in the UK favor products of enterprises which actively reduce environmental pollution. Due to a heightened awareness of social responsibility and environmental protection, consumers increasingly prefer green and low-carbon products [2]. Therefore, the level of carbon-emissions in the production and operation process will affect the utility and value for consumers, and ultimately affect the market demand for products.

Once an enterprise adopts an environment-friendly strategy, its products should likely be favored by consumers and so its market share should increase [3–5]. The UK Carbon Trust shows

that consumers are willing to pay more money for some greener products [6]. According to the report of the current situation of China's public green consumption (2019 Edition), 83.34% of the respondents expressed support for green consumption behavior [7]. Therefore, the huge consumer market generated by consumers' preferences for green environmental protection products has great significance to enterprises.

Furthermore, with the aggravation of climate problems such as environmental pollution and the greenhouse effect, more and more enterprises are implementing low-carbon management of their supply chain, and more and more consumers are starting to prefer green products. Low carbon not only causes the renewal of products, but also changes the existing production or distribution mode, increases the cost of emission-reduction, and further affects emission-reduction measures and product-pricing strategies. In order to maintain the stability of the market, the government can offer various kinds of subsidies. Common subsidies include direct subsidies for low-carbon enterprises and subsidies for consumers who buy green products. However, for different government subsidies, the low-carbon behavior of consumers affects the operation decision of the supply chain. At the same time, pricing and emission-reduction decisions will be different under the different power structure.

Therefore, considering the low-carbon behavior of consumers, we study product-pricing and carbon-emission problems in a green supply chain composed of a manufacturer and a retailer with different government subsidies. We establish a Stackelberg game model with a different power structure for various government subsidies [8]. There are two kinds of government subsidy: subsidies for manufacturers which produce green products and subsidies for consumers who buy green products. A supply chain includes two kinds of power structures: the manufacturer leader and the retailer leader [9]. Thus, four kinds of supply chain subsidy structure models are formed, namely, a government-subsidized manufacturer where the manufacturer is the leader (MM), a government-subsidized manufacturer where the retailer is the leader (MR), government-subsidized consumers where the manufacturer is the leader (CM), and government-subsidized consumers where the retailer is the leader (CR). We analyze a version of the model, give the optimal pricing strategy and the optimal carbon-emission-reduction decision for different power structures under various government subsidies, and analyze the influence of a government product subsidy coefficient on the optimal decision results.

This study examined how to arrange the generation and pricing of supply chain members in the case of consumer green preference with different government subsidies.

(1) We built different decision models for one manufacturer and one retailer's green supply chain, studying the pricing and emission-reduction decisions of the supply chain with different power structures under two forms of government subsidies, and analyzed how the subsidy coefficient affects these decisions.

(2) Under different power structures, we analyzed the impacts of the subsidy coefficient and pricing strategy on the performance of green supply chain members and the whole system.

## 2. Literature Review

In this section, we review two aspects of the literature, supply chain decision-making for low-carbon behavior of consumers and supply chain research for different forms of government subsidies.

### 2.1. Supply Chain Decision for Low-Carbon Behavior of Consumers

In the context of global advocacy for reducing carbon, many scholars have studied the supply chain decision-making for low-carbon behavior of consumers, and made some breakthroughs. Studies by Laroche et al. and others show that more consumers are willing to support higher prices for green products [10]. When considering different environmental regulations, Letpath and Balakrishnan established two game models to help enterprises optimize the product mix and output decision [11]. Using a cross supply chain model, Ghosh et al. analyzed the impact of greening policies [12]. Considering the reference price effect of consumers, Martin-Herran et al. studied the pricing decision of the supply chain [13].

Next, combined with the environmental preferences of consumers, Liu et al. studied the decision-making behavior of manufacturer and retailer [14]. Zhao et al. analyzed the Stackelberg game situation in which the manufacturer dominated suppliers and the situation of vertical cooperation between manufacturer and suppliers in the supply chain with the help of a differential game [15]. Using a two-level supply chain decision-making model, Liu et al. improved the newsboy model, and studied the impact of government low carbon-emission-reduction price subsidy on suppliers as a leader [16]. By the establishment of centralized and decentralized game models, Xiong obtained the optimal strategy combination of manufacturer and retailer under the two decision-making modes, and analyzed how the impact of consumer environmental awareness affected retail price [17]. Using an integrated structure, decentralized structure, and repurchase contract structure, Zhang et al. constructed a single supply chain system composed of a manufacturer and a retailer, and analyzed the optimal decision of conventional products and green products [18].

Additionally, Wang et al. considered the government's carbon-emission tax policy in the supply chain system [19]. Comparing the centralized and decentralized structure, Li et al. found that the centralized structure was better for the supply chain and enterprises [20]. Ji et al. considered consumers' preference in a dual-channel supply chain [21]. In a low-carbon consumer environment, Wang et al. built a dual-channel supply chain model, discussed the pricing strategies and profits for the supply chain members in two cases, and found that improving consumers' low-carbon preference was more acceptable to the supply chain members [22].

## 2.2. Supply Chain Decision with Different Government Subsidies

The existing literature at home and abroad, which combines government subsidy policy and supply chain low-carbon policy, focuses on government subsidies for supply chain enterprises. For example, Yakita and Yamauchib analyzed the impact of environmental R&D spillovers for various power structures [23]. Li et al. analyzed the optimal cost input of enterprise emission-reduction and government subsidy based on a decentralized decision-making model without government subsidies [24]. Xu et al. constructed the differential pricing model of green products and no-green products with government subsidies from the perspective of reducing the cost of green products, and obtained the optimal pricing strategy [25]. With carbon concerned demand, Du et al. studied how the low-carbon supply policies affect supply chain performance [26].

From the perspective of government subsidies to consumers, Huang et al. analyzed the impact of government subsidies on the electric vehicle industry and the environment [27]. Yu et al. analyzed the green policy effect of optimal production with consumer environmental awareness [28]. Yang and Fu studied the policy effect on government subsidies to consumers and corresponding supply chain member enterprises [29]. Based on the analysis of the pricing and emission-reduction optimization strategy, a contract mechanism for supply chain coordination and optimization is designed by using the Robinson bar-buying model [30].

In China, Xia et al. analyzed the impact of government subsidies on Chinese consumers' perceptions and found that not all consumers are looking forward to government subsidies [31]. Cohen et al. analyzed the green subsidy policy of consumers by building a game model composed of government and suppliers [32]. To study the energy saving and emission-reduction in a supply chain, Yi and Li established a Stackelberg model of a retailer and a manufacturer, and found that a government subsidy can increase the profits of supply chain members [33,34]. Under the cap-and-trade policy and low-carbon subsidy policy, Cao et al. studied policies which are more conducive to the production decision-making of enterprises, and found that the profits of production enterprises are not entirely dependent on low-carbon subsidy policies [35]. Considering consumer low-carbon awareness, Xia et al. established a game model dominated by the manufacturer, demonstrating that it can effectively improve the incentive for supply chain members to invest in low-carbon industries and improve their investment profits [36].

To summarize, some literature only considered government subsidies for enterprises, and they did not make a horizontal comparison between government subsidies for manufacturers and those for consumers; other literature did not consider the different power structures or the various kinds of government subsidies.

## 3. Problem Description and Hypotheses

### 3.1. Basic Assumptions

We considered a two-level green supply chain of a manufacturer and a retailer under different power structures and different government subsidies. In this system, the manufacturer decides the wholesale price and carbon-emission level, the retailer sells the products to consumers, and the consumers have a low-carbon preference. Therefore, because of the low-carbon behavior of consumers, the market demand for products is not only related to their prices but also affected by the carbon-emission level [37,38]. According to two studies [39], the market demand of products is

$$D(p, e) = 1 - p + \gamma e \tag{1}$$

where $p$ is the retailer price of unit product, $\gamma$ is the impact of emission-reduction level (rate) on demand, and considering that consumers are more sensitive to price than emission-reduction level, suppose $0 < \gamma < 1$, $e$ is the carbon-emission level of unit product.

Manufacturers need to use funds to improve emission-reduction technology. In other words, the improvement of emission-reduction technology requires a large monetary investment. According to two studies [40,41], the cost of carbon-emission-reduction technology is

$$C(e) = \frac{ke^2}{2} \tag{2}$$

where, $k$ is the manufacturer's carbon-emission-reduction effort coefficient. In order to ensure that the emission-reduction cost has a convex function, let $k > 1$.

For the government, in order to better promote the development of low-carbon economy, energy conservation and emission-reduction in the production field, the government often adopts two forms of subsidies. One is direct subsidies for low-carbon product manufacturers; the other is subsidies for consumers who buy green products. According to Cohen et al. [32], the government subsidies for unit green products are $he$, where $h$ is the adjustment factor of the subsidy coefficient of the unit product.

### 3.2. Model Establishment

When the government subsidizes the manufacturer who produces green products, the expected profits of the members of the green supply chain respectively are,

$$\pi_M^m = (w + he)(1 - p + \gamma e) - \frac{1}{2}ke^2, \tag{3}$$

$$\pi_R^m = (p - w)(1 - p + \gamma e) \tag{4}$$

where $w$ is the wholesale price of unit product, and let $p > w$.

When the government subsidizes consumers who buy green products, the expected profits of the members of the green supply chain respectively are,

$$\pi_M^c = w(1 - p + \gamma e + he) - \frac{1}{2}ke^2 \tag{5}$$

$$\pi_R^c = (p - w)(1 - p + \gamma e + he) \tag{6}$$

## 4. Model Solution and Analysis

In a two-level green supply chain of a manufacturer and a retailer under different power structures and different government subsidies, the manufacturer can dominate the supply chain system, and the retailer can also dominate the supply chain system. They have a Stackelberg game, and they are risk neutral and complete information among members of supply chain.

### 4.1. Government Subsidized Manufacturer When Manufacturer Is the Leader

When the manufacturer is the leader in a green supply chain, it first determines its own decision variables, and then the retailer determines its decision variable.

With the help of the backward induction method, the second derivative of retailer profit function to wholesale price is $\partial^2 \pi_R^{mm} / \partial p^{mm2} = -2 < 0$. The retailer's profit function is a concave function of the retail price, and the retail price can be obtained by its first-order condition. Retail price is a function of wholesale price and emission level. According to $\partial \pi_R^{mm} / \partial p^{mm} = 1 - 2p + \gamma e + w = 0$, then

$$p = \frac{1 + \gamma e + w}{2} \tag{7}$$

$$\pi_M^{mm} = \frac{1}{2}\left(w + w\gamma e - w^2 + he + h\gamma e^2 - hew - ke^2\right) \tag{8}$$

$$H = \begin{vmatrix} \frac{\partial^2 \pi_M^{mm}}{\partial w^2} & \frac{\partial^2 \pi_M^{mm}}{\partial w \partial e} \\ \frac{\partial^2 \pi_M^{mm}}{\partial e \partial w} & \frac{\partial^2 \pi_M^{mm}}{\partial e^2} \end{vmatrix} = \frac{4k - (\gamma + h)^2}{4} \tag{9}$$

By introducing the retail price into the profit function of the manufacturer, and using the properties of Hessian matrix, it is judged that when $0 < h < 2\sqrt{k} - \gamma$, the profit function of the manufacturer is a combined concave function of the emission level and the wholesale price, and we get the optimal wholesale price and emission level. Then, when we put the optimal wholesale price and the optimal emission level into the above retail price, we get the optimal retail price.

### 4.2. Government Subsidized Manufacturer When Retailer Is the Leader

When the retailer is the leader in a green supply chain, it determines its own decision variable first, and then the manufacturer determines the decision variables.

Let $p = w + t$, where $t$ is the price earning of unit product, and let $t > 0$. With the help of a Hessian matrix property, when $0 < h < \sqrt{2k} - \gamma$, the profit function of the manufacturer is a combined concave function about the wholesale price and the emission level, and then we can get the wholesale price and the optimal emission level. Their functions are all related to $t$. Then, we put the optimal wholesale price and the optimal emission level into the profit function of the retailer, to find the first derivative of the parameter variable $t$. Finally, we can get the optimal wholesale price, the optimal emission level, and the optimal retail price. In this way, when the government subsidizes manufacturer, we can get the optimal decisions of green supply chain in Table 1.

**Table 1.** The decision results of two different power structures in the case of government subsidies to manufacturers.

| | $w^*$ | $e^*$ | $p^*$ |
|---|---|---|---|
| MM | $\dfrac{h+\gamma}{4k-(\gamma+h)^2}$ | $\dfrac{2k-h^2-h\gamma}{4k-(\gamma+h)^2}$ | $\dfrac{\left(4k-(\gamma+h)^2\right)+(h+\gamma)(1-h\gamma)+2k\gamma}{2\left(4k-(\gamma+h)^2\right)}$ |
| MR | $\dfrac{h+\gamma}{2\left(2k-(\gamma+h)^2\right)}$ | $\dfrac{k-h^2-\gamma h}{2\left(2k-(\gamma+h)^2\right)}$ | $\dfrac{h+\gamma+2k-(\gamma+h)^2}{2\left(2k-(\gamma+h)^2\right)}$ |

### 4.3. Government Subsidized Consumers When Manufacturer Is the Leader

When the manufacturer is the leader in a green supply chain, it first determines its own decision variables, and then the retailer determines its own decision variable according to the decision of the manufacturer. When $0 < h < 2\sqrt{k} - \gamma$, the profit function of the manufacturer is a combined concave function about wholesale price and emission level.

### 4.4. Government Subsidized Consumers When Retailer Is the Leader

When the retailer is the leader in a green supply chain, the retailer determines its own decision variable first, and then the manufacturer determines its own decision variables according to the decision-makers of the retailer. When $0 < h < \frac{1}{2}\sqrt{k^3 + 6k + 1} + \frac{1}{2}(k-1) - \gamma$, the profit function of the manufacturer is a combined concave function about wholesale price and emission level. In this way, when the government subsidizes consumers, we can get the optimal decisions of green supply chain in Table 2.

**Table 2.** The decision results of two different power structures when the government subsidizes consumers.

| | $w^*$ | $e^*$ | $p^*$ |
|---|---|---|---|
| CM | $\dfrac{h+\gamma}{4k-(\gamma+h)^2}$ | $\dfrac{2k}{4k-(\gamma+h)^2}$ | $\dfrac{\left(4k-(\gamma+h)^2\right)+(h+\gamma)(2k+1)}{2\left(4k-(\gamma+h)^2\right)}$ |
| CR | $\dfrac{h+\gamma}{2\left(2k-(\gamma+h)^2\right)}$ | $\dfrac{k}{2\left(2k-(\gamma+h)^2\right)}$ | $\dfrac{h+\gamma+2k-(\gamma+h)^2}{2\left(2k-(\gamma+h)^2\right)}$ |

## 5. Comparison of Decision Analysis

In order to prove conveniently, we compared the decision results of each model under the limited condition $0 < h < \frac{1}{2}\sqrt{\gamma^3 - \gamma^2 - \gamma + 4\gamma k + 8k + 1} - (1+\gamma) < \sqrt{2k} - \gamma$.

**Theorem 1.** *Based on the different power structure of two government subsidies, the wholesale price has the following relationship:*

*(1) When the government subsidizes the manufacturer who produces green products, then $w^{mr*} > w^{mm*}$.*

*(2) When the government subsidizes consumers who buy green products, then $w^{cr*} > w^{cm*}$.*

**Proof.** (1) When the government subsidizes the manufacturer who produces green products, we can see the value of wholesale price in Table 1, where $w^{mm*} - w^{mr*} = \dfrac{h+\gamma}{4k-(\gamma+h)^2} - \dfrac{h+\gamma}{2\left(2k-(\gamma+h)^2\right)} = -h < 0$. The (1) of Theorem 1 is proofed.

(2) When the government subsidizes consumers who buy green products, we can see the value of wholesale price in Table 2, where $w^{cm*} - w^{cr*} = \dfrac{h+\gamma}{4k-(\gamma+h)^2} - \dfrac{h+\gamma}{2\left(2k-(\gamma+h)^2\right)} = -h < 0$. The (2) of Theorem 1 is proofed.

Theorem 1 is proofed. □

Theorem 1 shows that the wholesale price under the power structure of a retailer is always the largest, which is always higher than that under the power structure of a manufacturer, no matter whether the government directly subsidizes the manufacturer of green products or subsidizes the consumers who buy green products.

**Theorem 2.** *Based on the different power structure of two government subsidies, the carbon-emission level has the following relationship:*

*(1) When the government subsidizes the manufacturer who produces green products, then $e^{mm*} > e^{mr*}$.*

*(2) When the government subsidizes consumers who buy green products, then $e^{cm*} > e^{cr*}$.*

**Proof.** We can follow the proof idea of Theorem 1 to prove Theorem 2. □

Theorem 2 shows that no matter whether the government directly subsidizes the low-carbon product manufacturer or the consumers who buy the green products, the carbon-emission level is always the largest when the manufacturer is the power structure, which is always higher than the retail price when the retailer is the power structure. Once the manufacturer dominates the supply chain structure, it will increase the carbon-emission level of products, because it believes that the government will definitely subsidize, whether directly to manufacturers or to consumers. If the government directly subsidizes the manufacturer, the manufacturer can reduce the cost of carbon-emissions; if the government subsidizes the consumers, the manufacturer will raise the level of carbon-emissions while raising the price of products, and finally, the carbon subsidies given by the government are transferred to the manufacturer.

**Theorem 3.** *Based on the different power structure of two government subsidies, the retail price has the following relationship:*

    *(1) When the government subsidizes the manufacturer who produces green products, then $p^{mm*} > p^{mr*}$.*

    *(2) When the government subsidizes consumers who buy green products, then $p^{cm*} > p^{cr*}$.*

**Proof.** We can follow the proof idea of Theorem 1 to prove Theorem 3. □

Theorem 3 shows that no matter whether the government directly subsidizes the manufacturers of green products or the consumers who buy green products, the retail price is always the largest when the manufacturer is the power structure, which is always higher than the retail price when the retailer is the power structure.

So, we can get the following three lemmas.

**Lemma 1.** *No matter whether the government directly subsidizes the manufacturer for green products or the consumers who buy green products, the wholesale price of the manufacturer is directly proportional for the emission-reduction subsidy coefficient under the two power structures.*

Lemma 1 shows that in the environment of consumers' low-carbon behavior, as long as the government subsidizes low-carbon product manufacturers or consumers who buy green products, the wholesale price determined by low-carbon product manufacturers has nothing to do with the power structure of the supply chain, but only has to do with the number of government subsidies for carbon-emission. The increase of lattice is bound to be detrimental to the enthusiasm of consumers. Therefore, the government needs to set a reasonable range of subsidy coefficient, ensuring that manufacturers are willing to participate in manufacturing green products while protecting the interests of consumers.

**Lemma 2.** *When the government directly subsidizes the manufacturer of green products, the carbon-emission level of the manufacturer and the retail price of the retailer are inversely proportional to the emission-reduction subsidy coefficient in the case of the manufacturer's power structure; the carbon-emission level of the manufacturer and the retail price of the retailer are all directly proportional to the emission-reduction subsidy coefficient in the case of the retailer's power structure.*

Lemma 2 shows that in the environment of consumers' low-carbon behavior, when the government directly subsidizes low-carbon product manufacturers, the carbon-emission level of manufacturers and the retail price of retailers are not only related to the carbon-emission-reduction subsidy coefficient, but also related to the power structure of the supply chain. In the case of a manufacturer's power structure, the larger the carbon-emission subsidy coefficient, the smaller the manufacturer's carbon-emission level and retailer's retail price; on the contrary, in the case of a retailer's power structure, the larger the carbon-emission subsidy coefficient, the greater the manufacturer's carbon-emission level and retailer's retail price. When the government gives direct subsidies to the manufacturer, the larger the

subsidy coefficient is, the more likely consumers are to be in the power structure of the manufacturer. In this way, consumers' low-carbon preference can buy products with low carbon-emission levels and at a reasonable price.

**Lemma 3.** *When the government subsidizes the consumers who buy green products, the manufacturer's carbon-emission level and the retailer's retail price are directly proportional to the emission-reduction subsidy coefficient under the two power structures.*

Lemma 3 shows that in the environment of consumers' low-carbon behavior, when the government subsidizes the consumers who buy green products, the carbon-emission level of the manufacturer and the retail price are only related to the carbon-emission-reduction subsidy coefficient, but not to the power structure. When the government gives subsidies to consumers, the larger the subsidy coefficient is, the higher the carbon-emission level of the manufacturer and the retail price of retailer are. Therefore, consumers hope that the government will not subsidize consumers, which will lead to the increase of carbon-emission level of the manufacturer and the increase of the retail price of products, which is not conducive to the development of a low-carbon economy and society. In other words, consumers hope that the government can directly give low-carbon product manufacturer direct subsidies. This effect will be obvious, which is conducive to the reduction of carbon-emission levels and retail price.

## 6. Numerical Analysis and Discussion

We further verified the validity of the above models and theorems and made the calculation process simple and convenient. Assuming that the manufacturer's carbon-emission-reduction cost coefficient is $k = 20$, and the impact of carbon product emission-reduction level (rate) on demand is $r = 0.6$, combined with the range of government subsidy coefficient in the analysis, the government subsidy coefficient $h$ changes between 0.2 and 4.2. Through calculation, the impact of the change of the government subsidy coefficient on the decision results is shown in Tables 3–5.

Table 3 shows that the wholesale prices of manufacturers have not changed under the same dominant power structure, regardless of the subsidy method adopted by the government. Under the same government subsidy coefficient, the wholesale price of manufacturers has nothing to do with the type of government subsidy, only with the dominant power structure of the supply chain. Under a different government subsidy coefficient, the wholesale price of manufacturers increases with the increase of the subsidy coefficient. When the government directly subsidizes the manufacturer, the wholesale price under the dominant power structure of the retailer is always the largest, and always greater than the wholesale price under the dominant power structure of the manufacturer; when the government subsidizes the consumer, the wholesale price under the dominant power structure of the retailer is always the largest, and always greater than the wholesale price under the dominant power structure of the manufacturer, and the same dominant power structure. Therefore, different government subsidies have no effect on the wholesale price of manufacturers, that is $w^{mr*} = w^{cr*} > w^{mm*} = w^{cm*}$.

**Table 3.** The influence of subsidy coefficient on the wholesale price under different power structures.

| | Government Subsidized Manufacturers | | Government Subsidized Consumers | |
| | MM | MR | CM | CR |
| --- | --- | --- | --- | --- |
| $h$ | $w^{mm*}$ | $w^{mr*}$ | $w^{cm*}$ | $w^{cr*}$ |
| 0.6 | 0.0153 | 0.0156 | 0.0153 | 0.0156 |
| 1.2 | 0.0234 | 0.0245 | 0.0234 | 0.0245 |
| 1.8 | 0.0323 | 0.0350 | 0.0323 | 0.0350 |
| 2.4 | 0.0423 | 0.0484 | 0.0423 | 0.0484 |
| 3.0 | 0.0537 | 0.0666 | 0.0537 | 0.0666 |
| 3.6 | 0.0674 | 0.0939 | 0.0674 | 0.0939 |
| 4.2 | 0.0843 | 0.1415 | 0.0843 | 0.1415 |

Table 4 shows that no matter what kind of subsidy is adopted by the government, under the same dominant power structure, the manufacturer's carbon-emission level does not change much; on the contrary, under the same government subsidy, under different dominant power structures, the manufacturer's carbon-emission level changes greatly, which is always that the manufacturer's carbon-emission level under the dominant power structure is higher than that under the retailer's dominant power structure emission level. When the government directly subsidizes the manufacturer, the carbon-emission level under the dominant power structure of the manufacturer is always the largest, always greater than that under the dominant power structure of the retailer; when the government subsidizes the consumer, the carbon-emission level under the dominant power structure of the manufacturer is always the largest; also always greater than that under the dominant power structure of the retailer, that is, $e^{cm*} > e^{mm*} > e^{cr*} > e^{mr*}$.

**Table 4.** The influence of subsidy coefficient on the emission-reduction level of unit product under different power structures.

| | Government Subsidized Manufacturers | | Government Subsidized Consumers | |
|---|---|---|---|---|
| | **MM** | **MR** | **CM** | **CR** |
| $h$ | $e^{mm*}$ | $e^{mr*}$ | $e^{cm*}$ | $e^{cr*}$ |
| 0.6 | 0.5000 | 0.2500 | 0.5092 | 0.2593 |
| 1.2 | 0.4930 | 0.2427 | 0.5211 | 0.2720 |
| 1.8 | 0.4806 | 0.2290 | 0.5388 | 0.2921 |
| 2.4 | 0.4620 | 0.2065 | 0.5634 | 0.3226 |
| 3.0 | 0.4356 | 0.1701 | 0.5967 | 0.3698 |
| 3.6 | 0.3990 | 0.1091 | 0.6414 | 0.4472 |
| 4.2 | 0.3483 | −0.0047 | 0.7022 | 0.5896 |

Table 5 shows that no matter what kind of subsidy is adopted by the government, the wholesale price of retailers changes greatly under the same dominant power structure; on the contrary, under the same government subsidy and different dominant power structures, the wholesale price of retailers under the manufacturer's dominant power structure is the largest, which is always higher than that under the retailer's dominant power structure. When the government directly subsidizes the manufacturer, the wholesale price of the retailer under the manufacturer's dominant power structure is always the largest, which is always greater than the wholesale price under the retailer's dominant power structure; when the government subsidizes the consumer, the wholesale price of the retailer under the manufacturer's dominant power structure is always the largest, which is always greater than the wholesale price under the retailer's dominant power structure; and with the retailer under the dominant power structure, the different government subsidies have no effect on the retailer's sales price, that is $p^{cm*} > p^{mm*} > p^{cr*} = p^{mr*}$.

**Table 5.** The influence of subsidy coefficient on the selling price under different power structures.

| | Government Subsidized Manufacturers | | Government Subsidized Consumers | |
|---|---|---|---|---|
| | **MM** | **MR** | **CM** | **CR** |
| $h$ | $p^{mm*}$ | $p^{mr*}$ | $p^{cm*}$ | $p^{cr*}$ |
| 0.6 | 0.6576 | 0.5156 | 0.8131 | 0.5156 |
| 1.2 | 0.6596 | 0.5245 | 0.9807 | 0.5245 |
| 1.8 | 0.6603 | 0.5350 | 1.1627 | 0.5350 |
| 2.4 | 0.6597 | 0.5484 | 1.3662 | 0.5484 |
| 3.0 | 0.6575 | 0.5666 | 1.6008 | 0.5666 |
| 3.6 | 0.6534 | 0.5939 | 1.8807 | 0.5939 |
| 4.2 | 0.6466 | 0.6415 | 2.2275 | 0.6415 |

## 7. Conclusions

In the context of low-carbon behavior of consumers, we constructed a green supply chain system composed of a manufacturer and a retailer and studied the pricing and emission-reduction decision-making of the supply chain with different power structures for two forms of government subsidies. Through solving the model and comparing the results of each model, the results show the following:

(1) Whether the government directly subsidizes the manufacturer of green products or the consumers who buy green products, the wholesale price of the manufacturer increases gradually with the increase of the emission-reduction subsidy coefficient under the two power structures, and the wholesale price of the manufacturer's decision-making is higher under the leading power structure of the retailer.

(2) In the case of the manufacturer's power structure, the government's direct subsidy to green product manufacturers is not as efficient as a direct subsidy to consumers; in the case of the retailer's power structure, the government's direct subsidy to green product manufacturers is better than its direct subsidy to consumers.

(3) In terms of consumer protection, the retail price of the retailer in the supply chain with the power structure of the retailer is the lowest. The government should choose a reasonable subsidy coefficient range. In terms of reducing environmental pollution, the level of carbon-emission reduction is the best under the power structure of the retailer in the form of government subsidies to manufacturers. Currently, the carbon-emission level of low-carbon product manufacturers is the lowest, which is conducive to the healthy development of a low-carbon economy and ecological environment.

The difference between this paper and the study in reference [3] is that the carbon-emission level is taken into account in the demand function. Not only does the price of green products affect the demand, but also the carbon-emission level of products affects the demand function. According to two studies [29,32], government subsidies consider not only direct subsidies to manufacturers but also direct subsidies to consumers, and further expand to different power structures to study the pricing and carbon-emission reduction of low-carbon supply chains, enriching and developing the research of supply chains.

The disadvantage is that the government only considers the subsidy coefficient, and does not participate in the supply chain decision-making and prediction markets [42]; therefore, the introduction of three-level supply chain research of government participation in decision-making will be one area of future research. In addition, how carbon trading, carbon tax, and other policies affect the pricing and supply chain operation of green products will be the focus of further research.

**Author Contributions:** The study is the result of full collaboration and therefore the authors accept full responsibility. The Sections 1, 2 and 7 are attributable to C.S.; the Sections 3–6 are attributable to X.L. and she is the corresponding author of our paper. W.D. is a postdoctoral fellow of Hehai University, and he checked all the mathematics in the paper. All authors have read and agreed to the published version of the manuscript.

**Funding:** This research was funds by the strategy analysis of competitive supply chain with power structure under network externalities, grant number 2017SJB0975, and the game analysis of supply chain with network externalities under different power structures, grant number 16XWR011.

**Conflicts of Interest:** The authors declare no conflict of interest.

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
