# Peer review of "Green Supply Chain Decisions Considering Consumers’ Low-Carbon Awareness under Different Government Subsidies"

_sustainability, doi:10.3390/su12062281_

Round 1
Reviewer 1 Report
The paper treats important aspects however it needs additional elaboration.
Authors should suggest some references especially for statements such as: ”With the enhancement of consumers’ sense of social responsibility and environmental protection awareness, consumers increasingly prefer green and low-carbon products.”
The 2016 China green consumer report might be treated as outdated since we live in 2020.
Stackelberg game model needs references.
More than 85% of terence’s are Chinese - Authors should take into consideration world-wide literature and accordingly are asked to develop literature review.
There are misspellings in surnames of researchers in the paper. Authors should also read again the paper very carefully, since it is full of typos, grammar mistakes, spelling errors etc.
Parameters w, t are not discussed with first use.
Hessian matrix should include several results which might be presented as well with their short discussion.
Results given in table 1 might be simplified.
Authors should check the mathematics in the paper before resubmission.
What are the sources of data given in tables? Are they real-life data or hypothetical ones? It should be described in the paper.
The Authors write very long, complex sentences - e.g. 11 lines long - this makes it very difficult to understand the content. Authors are asked to reconstruct such sentences.
The paper needs future research description. Prediction markets might be used for future research:
Czwajda L., Kosacka-Olejnik M., Kudelska I., Kostrzewski M., Sethanan K., Pitakaso R., 2019, Application of prediction markets phenomenon as decision support instrument in vehicle recycling sector, LogForum, Vol. 15, Issue 2, pp. 265-278. DOI: 10.17270/J.LOG.2019.329
Author Response
Comment 1.Authors should suggest some references especially for statements such as: ”With the enhancement of consumers’ sense of social responsibility and environmental protection awareness, consumers increasingly prefer green and low-carbon products.”
Response: Thanks for the suggestion. According to a survey for British Carbon Trust, more than two-thirds of people in the UK favor products of enterprises which actively reduce environmental pollution. Due to a heightened awareness of social responsibility and environmental protection, consumers increasingly prefer green and low-carbon products [2].
Comment 2.The 2016 China green consumer report might be treated as outdated since we live in 2020.
Response: We updated the data. According to the report the current situation of China's public green consumption (2019 Edition), 83.34% of the respondents expressed support for green consumption behavior. The purpose of our quotation is to show that Chinese consumers prefer more and more green products.
Comment 3.Stackelberg game model needs references.
Response: Thanks for the suggestion and providing the information. See reference 8 which we referenced.
Comment4.More than 85% of terence’s are Chinese - Authors should take into consideration world-wide literature and accordingly are asked to develop literature review.
Response: Thanks for the suggestion and providing the information. We have added a lot of English literature which related our research for the literature review. See the red letter for references.
Comment5. There are misspellings in surnames of researchers in the paper. Authors should also read again the paper very carefully, since it is full of typos, grammar mistakes, spelling errors etc.
Response: We have paid MDPI for English editing. The paper has undergone English language editing by MDPI. The text has been checked for correct use of grammar and common technical terms. See the red letter for modification.
Comment 6.Parameters w, t are not discussed with first use.
Response: Thanks for the suggestion. We discussed with first use.
(3)
(4)
Where w is the wholesale price of unit product, and let p> w.
Let p=w+t, where t is the price earning of unit product, and let t> 0.
Comment 7.Hessian matrix should include several results which might be presented as well with their short discussion.
Response: According to, then
(7)
(8)
(9)
By introducing the retail price into the profit function of the manufacturer, and using the properties of Hessian matrix, it is judged that when, the profit function of the manufacturer is a combined concave function of the emission level and the wholesale price
Comment 8.Results given in table 1 might be simplified.
Response: Thanks for the suggestion and providing the information. We simplified the Table 1, Table 2, and Theorem 1.
Table 1.The decision results of two different power structures in the case of government subsidies to manufacturers
|
|
|||
|
MM |
|||
|
MR |
Table 2.The decision results of two different power structures when the government subsidizes consumers
|
|
|||
|
CM |
|||
|
CR |
Theorem 1. .
Comment 9Authors should check the mathematics in the paper before resubmission.
Response: Thanks for the suggestion. We invited a post-doctor from Hohai University to join our team. He checked all the mathematical formulas in the paper and made some modifications. For example, he checked Table 1, Table 2, and Theorem 1.
Comment10. What are the sources of data given in tables? Are they real-life data or hypothetical ones? It should be described in the paper.
Response: Thanks. We further verified the validity of the above models and theorems, and made the calculation process simple and convenient. Assuming that the manufacturer’s carbon emission-reduction cost coefficient is k=20, and the impact of carbon product emission-reduction level (rate) on demand is = 0.6, combined with the range of government subsidy coefficient in the analysis, the government subsidy coefficient h changes between 0.2 and 4.2.
Comment 11.The Authors write very long, complex sentences - e.g. 11 lines long - this makes it very difficult to understand the content. Authors are asked to reconstruct such sentences.
Response: Thanks for the suggestion and providing the information. We changed and paid MDPI for English editing. This study examined how to arrange the generation and pricing of supply chain members in the case of consumer green preference with different government subsidies. The green supply chain comprises a manufacturer and a retailer, the government subsidizes manufacturer who produce green products and consumers who buy green products.
Comment 12.The paper needs future research description. Prediction markets might be used for future research: Czwajda L., Kosacka-Olejnik M., Kudelska I., Kostrzewski M., Sethanan K., Pitakaso R., 2019, Application of prediction markets phenomenon as decision support instrument in vehicle recycling sector, LogForum, Vol. 15, Issue 2, pp. 265-278. DOI: 10.17270/J.LOG.2019.329
Response: Thanks for the suggestion and providing the information. To describe the future research, we cited literature Czwajda et al(2019). See the red letter for references 42.

Reviewer 2 Report
A few minor notes:
1. The abstract should be structured as follows: goal - methods - tools - results - discussion (optional)
2. The introduction should clearly state the purpose of the article, the purpose of research and the purpose of the analysis. I did not find the clearly defined purpose of the article.
3. It is mandatory to make editorial and language corrections. It is absolutely necessary to delete the personal wording, i.e. "we study product pricing", "We build different decision", "we analyze the impacts of ..." and all others. The text should be written impersonally.
4. Section 3. is to speak of hypotheses. I didn't find them. If the hypothesis are (line 151-153) "Therefore, under the low-carbon behavior of consumers, the market demand of products is not only related to their prices, but also affected by the carbon
emission level "and (line 159-160)" Manufacturers need to use funds to improve emission reduction technology. In other words, the improvement of emission reduction technology needs to invest a lot of money ", unfortunately, they should be reformulated. In the current version they are theses, and they do not have to be proved by the analysis made by the authors. The hypotheses are supposed to contain the supposition.
5. The developed model is correctly exemplified and free of logical errors.
6. The literature review section should be supplemented with international (world) achievements. In its current form it is insufficient.
Good luck!
Author Response
Comment1. The abstract should be structured as follows: goal - methods - tools - results - discussion (optional)
Response: Thanks for the suggestion and providing the information. We reorganized the language according to your suggestion, and paid MDPI for English editing. See the abstract.
Comment2. The introduction should clearly state the purpose of the article, the purpose of research and the purpose of the analysis. I did not find the clearly defined purpose of the article.
Response: Thanks for the suggestion and providing the information. We gave the purpose of the article in 76 lines. This study examined how to arrange the generation and pricing of supply chain members in the case of consumer green preference with different government subsidies.
Comment3. It is mandatory to make editorial and language corrections. It is absolutely necessary to delete the personal wording, i.e. "we study product pricing", "We build different decision", "we analyze the impacts of ..." and all others. The text should be written impersonally.
Response: Thanks for the suggestion and providing the information. We have paid MDPI for English editing. The paper has undergone English language editing by MDPI. The text has been checked for correct use of grammar and common technical terms. See the red letter for modification.
Comment4. Section 3. is to speak of hypotheses. I didn't find them. If the hypothesis are (line 151-153) "Therefore, under the low-carbon behavior of consumers, the market demand of products is not only related to their prices, but also affected by the carbon
emission level "and (line 159-160)" Manufacturers need to use funds to improve emission reduction technology. In other words, the improvement of emission reduction technology needs to invest a lot of money ", unfortunately, they should be reformulated. In the current version they are theses, and they do not have to be proved by the analysis made by the authors. The hypotheses are supposed to contain the supposition.
Response: Thanks for the suggestion and providing the information. We read this for other literatures.
Therefore, because of the low-carbon behavior of consumers, the market demand for products is not only related to their prices but also affected by the carbon-emission level [37,38]. According to two studies [39], the market demand of products is
(1)
Manufacturers need to use funds to improve emission-reduction technology. In other words, the improvement of emission-reduction technology requires a large monetary investment. According to two studies [40,41], the cost of carbon-emission-reduction technology is
(2)
Comment 5. The developed model is correctly exemplified and free of logical errors.
Response: Thanks for the suggestion. We invited a post-doctor from Hohai University to join our team. He checked all the mathematical formulas in the paper.
Comment 6. The literature review section should be supplemented with international (world) achievements. In its current form it is insufficient.
Response: Thanks for the suggestion and providing the information. We have added a lot of English literature which related our research for the literature review. See the red letter for references.

Reviewer 3 Report
- In Introduction would be very useful explain supply chain: from manufacturer up to consumer, because in Article no explain transportation and storage parts, which are important in green supply chain. For some goods main part of the green part of supply chain is logistics part.
- In Introduction Sources [1] and [2] did not link directly with the statements, can be recommend improve the text that more closely link with used Sources.
- In sub-sections 3.1, 3.2, 4.2 did not explain some formulas elements, for example (e), (w), (t) and (t ) differences and so on.
- Sub-sections 4.2, 4.3, 4,4 are very short and have not some resulting text or conclusions, mentioned sub-sections could be joint.
- In Article would be very useful case study section, which clarified main Authors ideas.
- Conclusions very long and not concrete, recommend improve.
Author Response
Comment1. In Introduction would be very useful explain supply chain: from manufacturer up to consumer, because in Article no explain transportation and storage parts, which are important in green supply chain. For some goods main part of the green part of supply chain is logistics part.
Response: The transportation and storage parts are important in green supply chain. But, this study examined how to arrange the generation and pricing of supply chain members in the case of consumer green preference with different government subsidies. Therefore, in this paper, we assume that the cost of the transportation and storage parts are zero. Our next article will consider transportation, inventory and other cost issues. This is a good suggestion. Thank you.
Comment2. In Introduction Sources [1] and [2] did not link directly with the statements, can be recommend improve the text that more closely link with used Sources.
Response: Thanks for the suggestion and providing the information. We added “According to a survey for the UK Carbon Trust,”,and paid MDPI for English editing. We think it will help us a lot.
Comment3. In sub-sections 3.1, 3.2, 4.2 did not explain some formulas elements, for example (e), (w), (t) and (t ) differences and so on.
Response: Thanks for the suggestion and providing the information. We explained some formulas elements. For example,
Where p is the retailer price of unit product, is the impact of emission-reduction level (rate) on demand, and considering that consumers are more sensitive to price than emission-reduction level, suppose, e is the carbon-emission level of unit product.
Where, k is the manufacturer’s carbon-emission-reduction effort coefficient. In order to ensure that the emission-reduction cost has a convex function, let k > 1.
Where w is the wholesale price of unit product, and let p> w.
Let p=w+t, where t is the price earning of unit product, and let t> 0.
Comment4. Sub-sections 4.2, 4.3, 4,4 are very short and have not some resulting text or conclusions, mentioned sub-sections could be joint.
Response: Thank you. Sub-sections 4.1 is simplified.
According to, then
(7)
(8)
(9)
But Sub-sections 4.2, 4.3, 4,4 are simplified the calculation process.
They are four kinds of supply chain subsidy structure models, namely, a government-subsidized manufacturer where the manufacturer is the leader (MM), a government-subsidized manufacturer where the retailer is the leader (MR), government-subsidized consumers where the manufacturer is the leader (CM), and government-subsidized consumers where the retailer is the leader (CR).
Comment5. In Article would be very useful case study section, which clarified main Authors ideas.
Response: Thanks. The case study section was verified the validity of the above models and theorems, and made the calculation process simple and convenient.
Comment6. Conclusions very long and not concrete, recommend improve.
Response: Thanks for the suggestion and providing the information. We improved the conclusions for three points. The conclusion is more concise and targeted. See the red letter for conclusions.

Round 2
Reviewer 1 Report
Authors took the effort of verifying all the paper. Changes in the paper can be accepted.
Reviewer 2 Report
The authors took all comments into account. I have no more comments.